# Toward a Clearer Characterization of Neuro-Symbolic Frameworks: A Brief Comparative Analysis

**Sania Sinha**                                               SINHASA3@MSU.EDU
**Tanawan Premsri**                                          PREMSRIT@MSU.EDU
**Parisa Kordjamshidi**                                       KORDJAMS@MSU.EDU
*Department of Computer Science and Engineering, Michigan State University*

**Editors:** Leilani H. Gilpin, Eleonora Giunchiglia, Pascal Hitzler, and Emile van Krieken

## Abstract

Neurosymbolic (NeSy) frameworks combine neural representations and learning with symbolic representations and reasoning. Combining the reasoning capacities, explainability, and interpretability of symbolic processing with the flexibility and power of neural computing allows us to solve complex problems with more reliability while being data-efficient. However, this recently growing topic poses a challenge to developers with its learning curve, lack of user-friendly tools, libraries, and unifying frameworks. In this paper, we characterize the technical facets of existing NeSy frameworks, such as the symbolic representation language, integration with neural models, and the underlying algorithms. A majority of the NeSy research focuses on algorithms instead of providing generic frameworks for declarative problem specification to leverage problem solving. To highlight the key aspects of Neurosymbolic modeling, we showcase three generic NeSy frameworks - *DeepProbLog*, *Scallop*, and *DomiKnowS*. We identify the challenges within each facet that lay the foundation for identifying the expressivity of each framework in solving a variety of problems. Building on this foundation, we aim to spark transformative action and encourage the community to rethink this problem in novel ways.

**Keywords:** Neurosymbolic, Comparing NeSy frameworks, DomiKnowS, DeepProbLog, Scallop, Combining learning and reasoning

## 1. Introduction

**Symbolic or *good old-fashioned* AI** focused on creating rule-based reasoning systems (Hayes-Roth, 1985) exemplified with early works such as the Physical Symbol System (Augusto, 2021; Newell, 1980) and ELIZA (Weizenbaum, 1966). However, drawbacks such as limited scalability due to the need to explicitly define domain predicates and rules for each task, lack of robustness in handling messy real-world data, and low computational efficiency led to a decline in the popularity of this paradigm, shifting the focus toward neural computing and deep learning. **Deep Learning** (LeCun et al., 2015; Ahmad et al., 2019) revolutionized AI as nuanced relationships in data could be learned by backpropagation through multiple layers of processing and creating abstract representations of data. However, it led to a loss of explainability (Li et al., 2023a), dependence on large amounts of data, and rising concerns about its environmental sustainability (Bender et al., 2021). **Neurosymbolic AI** (Hitzler and Sarker, 2022; Bhuyan et al., 2024), a combination of symbolic AI and reasoning with neural networks, attempts to incorporate the capabilities of both worlds and create systems that are data and time efficient, generalizable, and explainable.

Neurosymbolic models have been applied to several real-world applications (Bouneffouf and Aggarwal, 2022) in safety-critical areas (Lu et al., 2024) such as healthcare (Hossain and Chen, 2025) and autonomous driving (Sun et al., 2021). Several techniques have been proposed for this integration (Kautz, 2022; Jayasingha et al., 2025), trying to combine the pros and mitigate the cons from both symbolic and neural methods. However, due to lack of unified libraries to facilitate this research and the focus on specific algorithms rather than generic frameworks, this research becomes less impactful. Moreover, the few generic frameworks tend to vary in problem formulation, implementation, algorithms, and flexibility of use. This poses a challenge in being able to compare their performance uniformly or identify a research direction that improves on previous work. To alleviate this issue, we provide a comparative study with the following key contributions.

a) Identifying the main components of existing NeSy frameworks, b) Comparison of frameworks across the identified facets, c) Highlighting the requirements for the next generation of NeSy frameworks, building upon the drawbacks of the current systems and the possible interplays between the neural and symbolic components. We plan to expand this study to cover more frameworks while the three selected ones are used to explain the aspects of our characterization. These frameworks are demonstrated with an example task detailed in Appendix A, tying the comparative facets concretely with a technical implementation.[1] The MNIST Sum is a modified version of the classic MNIST digit recognition task (Lecun et al., 1998), where a model is given images of two digits and asked to predict their sum.

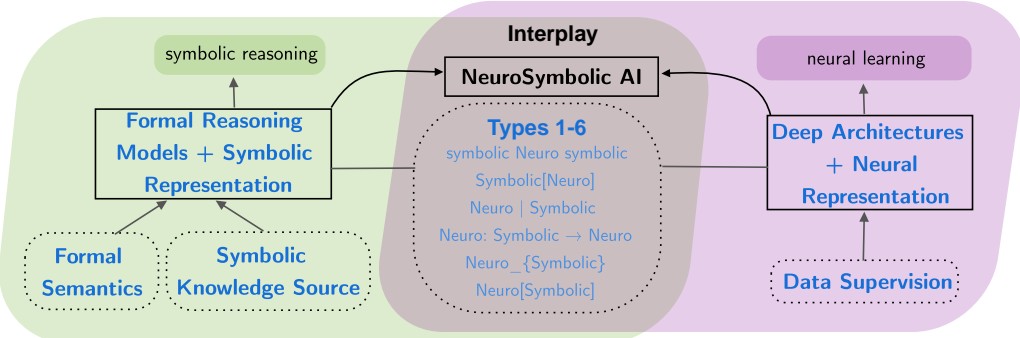

Figure 1: An overview of the main components of a neurosymbolic framework.

## 2. Neurosymbolic Frameworks

A NeSy framework should provide flexibility for modeling both neural and symbolic components (Kordjamshidi et al., 2016, 2015) and their interplay in a unified declarative framework, going beyond specific underlying algorithms and techniques. On the symbolic side, a generic framework should support a symbolic representation language that can be seamlessly connected to neural components and cover different symbolic reasoning mechanisms. On the neural side, we need to have the flexibility of connecting to various architectures, including various loss functions, sources of supervision, and training paradigms. More importantly, a

---

1. https://github.com/HLR/nesy-examples

| Framework | Symbolic | | Model Dec | Interplay | | LLM |
| | Lang | Knowledge Rep | | Algorithm | Eff | |
|---|---|---|---|---|---|---|
| CCN+ | None | Propositional Logic Clauses | ✗ | ReqL & ReqLoss | ✗ | ✗ |
| DomiKnowS | None | Concepts, Constraints | ✓ | Primal-Dual, Sampling Loss | ✗ | Faghihi et al. (2024) |
| DeepProbLog | ProbLog | Facts, Rules, Predicates | ✗ | Entailment | ✗ | ✗ |
| LEFT | None | First Order Logic | ✗ | Differentiable Reasoning | ✗ | Hsu et al. (2023) |
| PyReason | None | Constants, Relations, Facts, Rules | ✗ | Reasoning over graph | ✓ | ✗ |
| Scallop | DataLog | Rules, Relations | ✗ | Differentiable Reasoning | ✓ | Li et al. (2024) |

Table 1: Frameworks with their comparative factors. Lang: External language required, Knowledge Rep: Knowledge Representation, Model Dec: Model Declaration flexibility, Algorithm: Supported algorithm(s) for learning and inference, Eff: Computational efficiency considerations, LLM: Use of Large Language Models.

NeSy framework should provide a modeling language for specification and seamless integration of the two components in building pipelines or arbitrary composition of models. Such a NeSy framework should support neuro-symbolic training and inference beyond specific integration algorithms. We distinguish between *NeSy techniques* and *NeSy frameworks*. By techniques, we mean when task-specific solutions are provided (Lample and Charton, 2020; Burattini et al., 2002). For example, **AlphaGo** (Silver et al., 2016) introduced a reinforcement learning solution to Go, using Monte Carlo Tree Search as a symbolic component inside a neural network. Another example is **NS-CL** (Mao et al., 2019) (Neuro-Symbolic Concept Learner) that integrates neural perception with symbolic reasoning to learn visual concepts and compositional language grounding for VQA tasks. Many other techniques and algorithms are proposed for the interplay between the two paradigms (Badreddine et al., 2022; Cohen et al., 2017; Smolensky et al., 2016; Lima et al., 2005; Sathasivam, 2011; Serafini and d'Avila Garcez, 2016; Lamb et al., 2021) such as Inference Masked Loss (Guo et al., 2020), Semantic Loss (Xu et al., 2018), Primal-Dual (Nandwani et al., 2019), etc., later discussed in Section 6. They often lack the generality of frameworks, which are designed as broader tools intended for practical use and extensibility with new integration algorithms and with the capability of programming and configuring the two parts and their interplay.

In this work, we focus on a selection of generic NeSy frameworks. The following are examples of research efforts towards advancing the development of such general-purpose frameworks: **DeepProbLog** (Manhaeve et al., 2021) is a probabilistic logic programming language, incorporating neural predicates in logic programming with an underlying differentiable translation of logical reasoning. The probabilistic logic programming component is built on top of ProbLog (De Raedt et al., 2007). **DomiKnowS** (Rajaby Faghihi et al., 2021; Faghihi et al., 2023, 2024) is a declarative learning-based programming framework (Kord-

jamshidi et al., 2019) that integrates symbolic domain knowledge into deep learning. It is a Python framework, facilitating the incorporation of logical constraints that represent domain knowledge with neural learning in PyTorch. **Scallop** (Huang et al., 2021; Li et al., 2023b, 2024) is a framework that includes flexible symbolic representation based on relational data modeling, using a declarative logic programming built on top of Datalog (Abiteboul et al., 1995) with a framework for automatic differentiable reasoning. **LEFT** (Hsu et al., 2023) is a less generic framework designed for grounding language in visual modality and compositional reasoning over concepts. The framework consists of an LLM interpreter that converts natural language to logical programs. The generated programs are directed to a differentiable, domain-independent, and soft first-order logic-based executor. LEFT is limited to tasks requiring grounding language in vision such as visual question answering (Johnson et al., 2017; Yi et al., 2018; Liu et al., 2019). Building on this foundation, NeSyCoCo (Kamali et al., 2025) was introduced to address its limitations, particularly its struggle with lexical variety and handling unseen concepts. NeSyCoCo extends LEFT's approach by using distributed word representations to connect a variety of linguistically motivated predicates to neural modules, to alleviate reliance on a predefined predicate vocabulary. **PyReason** (Aditya et al., 2023) is a library built to support reasoning on top of outputs from neural networks. The neural component produces outputs such as labels or concept scores. While the symbolic component does graph-based reasoning using logic rules declared over a graph structure. It can produce an explanation trace for inference and has a memory-efficient implementation. **PLoT** (Wong et al., 2023) (Probabilistic Language of Thought) is a *proposed* framework leveraging neural and probabilistic modeling for generative world modeling. It models thinking with probabilistic programs and meaning construction with neural programs. The goal is to provide a language-driven unified thinking interface. **CCN+** (Giunchiglia et al., 2024) is a framework that modifies the output layer of a neural network to make results compliant with requirements that can be expressed in propositional logic. A requirement layer, ReqL, is built on top of the neural network. The standard cross-entropy loss is adapted into a ReqLoss to learn from the constraints in the ReqL layer. We characterize frameworks by: a) Symbolic knowledge representation language, b) Representation and flexibility of Neural Modeling, c) Model Declaration, d) Interplay between symbolic and sub-symbolic systems, and e) Usage of LLMs. Figure 1 shows the relationship between these aspects. The neural representations and the symbolic representations are the two main components of a neurosymbolic framework. The neural representation guides learning and obtaining supervision from the data, while the symbolic representations leverage symbolic reasoning, where the symbolic knowledge can be exploited during training or inference. Table 1 shows an overview of the frameworks across chosen features. For future sections, we focus on **DomiKnowS**, **DeepProbLog**, and **Scallop** to provide a deeper investigation of the challenges in each component. Due to differences in implementation, each framework allows for easy implementation of different tasks. With the chosen frameworks, we can solve the same task across all.

## 3. Symbolic Knowledge Representation

Generic Neuro-Symbolic (NeSy) systems and frameworks use symbolic knowledge representation languages to encode constraints, facts, probabilities, and rules. Frameworks vary in how they represent and integrate this symbolic knowledge. Many employ classical formal

logic, grounded in well-defined syntax and semantics, and adapt these representations and reasoning mechanisms within a unified integration framework. Some frameworks build on established formalisms such as logic programming or constraint satisfaction. In contrast, others take an entirely new hybrid semantics, while preserving conventional symbolic syntax. Figure 2 compares the implementation of symbolic knowledge (concepts or facts) for the MNIST Sum task. In general, the domain knowledge consists of the two concepts of *digits* and the *sum*. As can be seen, Domiknows represents a part of symbolic domain knowledge as a graph $G(V, E)$, where the nodes are the concepts in the domain and the edges are the relationships between them. Each node can have properties. More complex knowledge beyond entities and relations is expressed with a pseudo first-order logical language with quantifiers designed in Python. DomiKnowS mostly interprets the symbolic knowledge as logical constraints, such as the implementation of sum_combinations in the given example. Unlike the other frameworks, DomiKnowS does not build on predefined formal semantics. It follows a FOL-like syntax for symbolic logical representations, making it independent of the formal semantics of an underlying formal language and allows more flexibility of representations and adaptation to underlying algorithms in the framework. DeepProbLog, on the other hand, utilizes logical predicates that are originally a part of the probabilistic logic programs (Ng and Subrahmanian, 1992) of ProbLog (De Raedt et al., 2007), for its symbolic representation. These neural predicates obtain their probability distributions from the underlying neural models. Probabilistic facts, neural facts, and neural annotated disjunctions (nAD) whose probabilities are supplied by the neural component of the program can be added.

Here, digit is a neural predicate as indicated by the use of nn(...). DeepProbLog follows the formal semantics of Prolog (Clocksin and Mellish, 2003), followed by ProbLog, its probabilistic extension. Scallop adopts a relational data model for symbolic knowledge representation (Kolaitis and Vardi, 1990). It is built on the syntax and formal semantics of Datalog and its probabilistic extensions, relaxing the exact semantics of ProbLog. It allows for the expression of common reasoning, such as aggregation, negation, and recursion. Similar to DeepProbLog, some of these predicates in the symbolic part obtain their probability distribution from neural models, such as digit_1 and digit_2. While ProbLog requires exhaustive search for computations, Datalog can use top-k results and exploit database optimizations, making Scallop algorithmically more time-efficient.

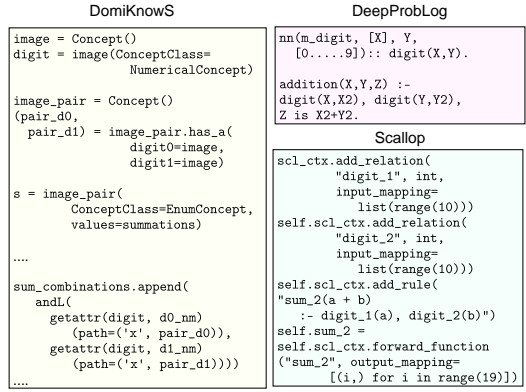

Figure 2: Comparison of Symbolic Representation across frameworks.

## 4. Neural Models Representations

The other core component of a NeSy system is the neural modeling that is integrated with the symbolic knowledge discussed above. The neural models are mostly wrapped up under the logical predicate names in most of the frameworks that have an explicit

logical knowledge representation language. To leverage the reasoning capabilities of the symbolic system available and the ability of neural models to learn abstract representations from data, the neural models are used as abstract concept learners for the concepts defined as logical predicates in the symbolic representation. The neural model representation is often used to predict probability distributions for the symbolic concepts based on raw sensory inputs. The neural modeling is often written using standard deep learning libraries, such as PyTorch (Paszke et al., 2019). Figure 3 shows snippets of neural modeling expressions across frameworks, highlighting differences in implementation. Scallop utilizes relatively standard neural modeling using PyTorch, while needing an added context of symbolic rules. Although integrated into Python, the context relation and rule setup are verbatim from Data-Log and only passed as a parameter to a function, which requires familiarity with DataLog and its semantics. DeepProbLog, on the other hand, needs manual configuration of the raw data and processing into queries, on top of other standard neural components. This processed data is passed into the neural network which is then connected to a ProbLog program, such as addition.pl in the figure. DomiKnowS's neural component is built in PyTorch. Unlike other frameworks, DomiKnowS has built-in components called *Readers, Sensors* and *Module learners* that make the connection to neural components and feeding data to them explicit in the program. This provides more flexibility in connecting the concepts to deterministic or probabilistic functions that can interact with other symbolic concepts. The module learner can also use custom models. This makes the interaction with raw data structured, transparent, and controllable.

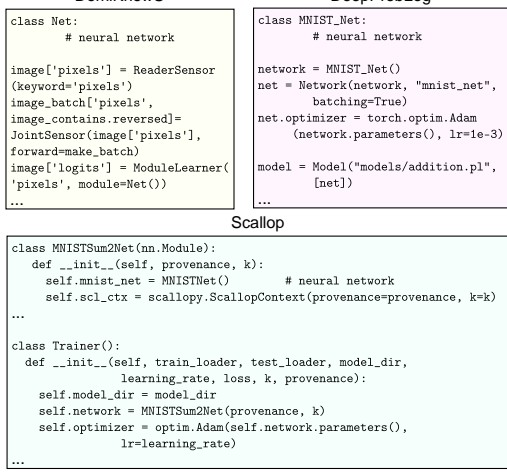

Figure 3: For neural integration, Domi-KnowS employs sensors and readers for reading in data, while a learner connects to a network. DeepProbLog connects the neural network to the ProbLog file, requiring data handling to construct the terms and queries from raw data. Scallop has an extra layer on top of the standard network that adds the symbolic context.

## 5. Model Declaration

Most frameworks utilize neural components as abstract concept learners and use a symbolic component to reason over the learned concepts. Each learner is a model and *model declaration* refers to the flexibility of modularizing and connecting different learners. Each learner can receive supervision independently. In most neurosymbolic frameworks, the supervision from data is usually provided based on the final output of the end-to-end model. For example, in an MNIST Sum task used throughout and detailed in Appendix A, the neural and symbolic components are trained based on the final output of the sum, without access

to individual digit labels in a semi-supervised setting. The task loss, e.g., a Cross-Entropy Loss, is computed, and errors are backpropagated through the differentiable operations that led to the output generation. For example, in DeepProbLog, we can declare a single loss function associated with the entire neural component. Gradient computations differ across frameworks depending on whether losses are defined individually for each neural output or specified as a single global loss function. However, there remains a need for models capable of incorporating supervision at multiple levels of their symbolic representations. In DomiKnowS, loss computation can be defined for each symbol. Since each concept is linked to both learning modules and ground-truth labels, their losses can be integrated seamlessly. This enables joint training of all concepts alongside the target task, allowing each concept to be optimized more effectively—leveraging available data without relying solely on the target task's output. In other words, it provides the flexibility of building pipelines of decision making, obtaining distant supervision in addition to joint training and inference.

## 6. Interplay between Symbolic and Sub-symbolic

Kautz (2022) provides a characterization of the possible interplays between symbolic and sub-symbolic components. This interplay of neurosymbolic can be explained by the concept of System 1 and System 2 thinking described in Kahneman (2011). Research in this field aims to create an ideal integration that seamlessly supports "thinking fast and slow" (Booch et al., 2021; Fabiano et al., 2023). Here, System 1 refers to the fast neural processing, while System 2 corresponds to the slower, more deliberate symbolic reasoning. *Different methods for the integration of symbolic reasoning and neural programming have been explored such as employing logical constraint satisfaction, integer linear programming, differentiable reasoning, probabilistic logic programming.* In this section, we will discuss a system-level algorithmic comparison of the different frameworks.

**DomiKnowS** models the inference as an integer linear programming problem to enforce the model to follow constraints expressed in first-order logical form (Van Hentenryck et al., 1992). The objective of the program is guided by the neural components, and the framework supports multiple training algorithms for learning from constraints. The Primal-Dual formulation (Nandwani et al., 2019) converts the constrained optimization problem into a min-max optimization with Lagrangian multipliers for each constraint, augmenting the original loss with a soft logic surrogate to minimize constraint violations. Sampling-Loss (Ahmed et al., 2022), inspired by semantic loss (Xu et al., 2018)and samples a set of assignments for each variable based on the probability distribution of the neural modules' output. Integer Linear Programming (ILP)(Cropper and Dumančić, 2022) formulates an optimization objective based on Inference-Masked Loss(Guo et al., 2020) to constrain the model during training. The training goal is to adjust the neural models to produce legitimate outputs that adhere to the given constraints. At prediction time, ILP can also be applied to enforce final predictions that comply with given constraints. DomiKnowS relies on the off-the-shelf optimization solver Gurobi (Gurobi Optimization, LLC, 2024) **DeepProbLog** models each problem as a probabilistic logical program that consists of neural facts, probabilistic facts, neural predicates, and a set of logical rules. A joint optimization of the parameters of the logic program is done alongside the parameters of the neural component. Neural network training is done using learning from entailment (Frazier and Pitt, 1993) while in ProbLog, gradient-based optimization is performed on the underlying

generated Arithmetic Circuits (Shpilka et al., 2010), which is a differentiable structure. The Arithmetic Circuits are transformed from a Sentential Decision Diagram (Darwiche, 2011) generated by ProbLog. Algebraic ProbLog (Kimmig et al., 2011) is used to compute the gradient alongside probabilities using semirings (Eisner, 2002). **Scallop** is similar in its setup to DeepProbLog where it creates an end-to-end differentiable framework combining a symbolic reasoning component with a neural modeling component. They aim to relax the formal semantics required by the use of ProbLog in DeepProbLog and instead rely on a symbolic reasoning language extending DataLog, built into their framework. They have a customizable provenance semiring framework (Green et al., 2007), where different provenance semirings, such as extended max-min semiring and top-k proofs semiring, allow learning using different types of heuristics for gradient calculations. Table 2 compares the computational efficiency of these models at training and inference time on a single training/testing example. As theoretically suggested, Scallop is expected to outperform other frameworks in inference and training speed, owing to its memory and time-efficient implementation in Rust. The results in Table 2 support this expectation, with Scallop achieving the fastest inference time, on par with DomiKnowS. In practice, DeepProbLog achieves slightly faster training performance than Scallop. This discrepancy may be due to overhead unrelated to the core algorithmic complexity. DomiKnowS exhibits slower training due to the overhead of uploading the entire graph of data into memory.

| Framework | Training Time(ms) | Inference Time(ms) |
|---|---|---|
| DomiKnowS | 37.72 | **2.34** |
| DeepProbLog | **5.84** | 3.24 |
| Scallop | 6.50 | **2.35** |

Table 2: Computation time in milliseconds (ms) for training and inference across frameworks based on one training/testing example.

## 7. Role of Large Language Models

Large foundation models hold significant promise for overcoming the bottleneck of acquiring symbolic representations, which are essential for symbolic reasoning and consequently in neurosymbolic frameworks. **Source of Symbolic Knowledge:** The symbolic knowledge in neuro-symbolic systems, which is integrated with the neural component, can originate from several distinct sources. While most systems require explicit, hand-crafted symbolic knowledge, earlier classical logic-based learning research can be used for automatically learning rules from data by using inductive logic programming (Nienhuys-Cheng and de Wolf, 1997; Bratko and Muggleton, 1995) or mining constraints. Nowadays, even LLMs can be utilized to generate symbolic knowledge (Pan et al., 2023a; Mirzaee and Kordjamshidi, 2023; Acharya et al., 2024; Xu et al., 2024a). Several neurosymbolic frameworks and systems have tried utilizing large foundation models to generate the symbolic knowledge, based on the task or query, to overcome the labor-intensive nature of hand-crafting rules for every single task and the time required in the automatic learning of symbolic knowledge from data (Ishay et al., 2023; Xu et al., 2024a; Yang et al., 2024). Extraction of symbolic representations from Foundation Models has become possible given the vast implicit knowledge stored within these models, such as LLMs and multimodal models, which are trained on massive and diverse corpora (Li et al., 2024; Petroni et al., 2019). These models can gen-

erate symbolic content (e.g., candidate rules, knowledge graph triples, or logic statements), perform reasoning that mimics symbolic inference, or act as components alongside symbolic modules (Fang and Yu, 2024). For example, LLMs can be prompted to extract facts from unstructured text, effectively populating a symbolic knowledge graph (Yao et al., 2025). Techniques like Symbolic Chain-of-Thought inject formal logic into the LLM's reasoning process, improving accuracy and explainability on logical reasoning tasks (Xu et al., 2024b). However, foundation models are prone to hallucinations and lack the strict logical guarantees of traditional symbolic systems (Zheng et al., 2024). Therefore, integrating foundation models often requires careful prompting, verification steps to ensure reliability (Xu et al., 2024b). **Generation of inputs to symbolic engines:** LLMs have also been used to generate translations from raw inputs, specially natural language, to symbolic language that is then fed into a symbolic reasoner. For example in Logic-LM (Pan et al., 2023b), LLMs are leveraged to convert a natural language query into symbolic language that is then solved by a symbolic reasoner. This method improves the performance of unfinetuned LLMs on logical reasoning-based tasks. DomiKnowS (Faghihi et al., 2024) takes this a step further by enabling users to describe problems in natural language which LLMs then use to generate relevant concepts and relationships. Through a user-interactive process, these concepts and relationships are refined iteratively. Finally, the LLM translates the user-defined constraints from natural language into first-order logic representations before converting them into DomiKnowS syntax. Some systems use LLMs in multiple capacities. In VIERA (Li et al., 2024), which is built on top of Scallop, 12 foundation models can be used as plugins. These models are treated as stateless functions with relational inputs and outputs. These foundation models can be either language models like GPT (OpenAI et al., 2024) and LLaMA (Touvron et al., 2023), vision models such as OWL-ViT (Minderer et al., 2022) and SAM (Kirillov et al., 2023), or multimodal models such as CLIP (Radford et al., 2021). These models can be used to extract facts, assign probabilities, or for classification, and are treated as "foreign predicates" in their interface. An older version, DSR-LM (Zhang et al., 2023) utilized BERT-based language models for perception and relation extraction, combined with a symbolic reasoner for question answering. LEFT uses LLMs both for generation of the concepts used for grounding and as an interpreter to generate the first-order logic program for a natural language query, that is solved by the symbolic executor.

## 8. Discussion and Future Direction

Table 1 summarizes the comparative aspects of existing frameworks and outlines future directions for optimizing, as we observe many columns marked with '✗', implying most frameworks present challenges that hinder the application and flexibility of the frameworks. While current frameworks are functional, future developments should take a more holistic approach that considers all aspects from an end-user perspective, aiming to improve usability as general-purpose libraries and foster wider adoption of neurosymbolic methods. **Symbolic Representation.** The generic neurosymbolic frameworks provide a formal knowledge representation language of their choice. The selected languages often are based on pure logical formalisms with established formal semantics, for example, Datalog or Prolog. However, we argue that knowledge representation for neurosymbolic frameworks needs to be an innovative language designed for this integration purpose with adaptable semantics with learning as the pivotal concept (Kordjamshidi et al., 2019). Restricting these frame-

works to classical AI formalisms and formal semantics limits the level of extension that can be made and restricts the support of various algorithms and types of integration.

**Neural Modeling.** Most of the examined frameworks leave neural modeling and the task of connecting the symbolic and sub-symbolic components to the user. This connection usually requires low-level data preprocessing, that's time consuming to implement. A lack of user-friendly libraries discourages developers from using neurosymbolic methods to solve downstream tasks. These frameworks need abstractions (Kordjamshidi et al., 2022) that improve user experience and remove the need to implement data processing from scratch.

**Model Declaration**. There is a need to be explicit about the low-level components of the neural architecture, enabling us to design interactions between neural and symbolic components and connect them as intended. The goal is to provide flexibility in designing arbitrary loss functions and connecting them to data for supervising concepts at various neural layers, which will allow any symbol to be learnable.

**Types of Interplay.** Considering Kautz (2022)'s classification, current frameworks are limited in supporting one or two ways of interactions. The "Algo" column in Table 1 shows that DeepProbLog and Scallop utilize one form of implementation, while DomiKnowS has multiple settings. One of the key challenges is determining the appropriate level of abstraction in a neural model after which reasoning should occur. The classification types demonstrate how a neural model can identify the relevant symbolic representations and suggest that neurosymbolic frameworks could leverage these models to learn and route inputs to the corresponding symbolic reasoning system. However, it remains unclear what level of abstraction is most effective for solving the end task in practice.

**LLM.** Drawbacks often associated with employing symbolic AI into neural computing, such as creation of the symbolic knowledge for integration, can be mediated with the use of LLMs and foundation models. LLMs have the potential to alleviate the classical issues in symbolic processing. Their vast knowledge can also be utilized to reduce the need for rebuilding neural components, allowing for flexible connections with different symbolic components.

## 9. Conclusion

Neurosymbolic AI presents a promising path forward in addressing the limitations of purely symbolic or neural approaches to AI. By integrating symbolic reasoning with neural learning, NeSy frameworks offer a balance between interpretability, data and time efficiency, and generalization. In this paper, we characterize the core components of NeSy frameworks and provide an analysis of some existing ones - DeepProbLog, Scallop, and DomiKnowS, illustrating the comparative facets. We identified some facets as symbolic knowledge and data representation, neural modeling, model declaration, method of integrating the symbolic and sub-symbolic systems, and role of LLMs. We identify key challenges in each facet that can guide us toward building the next generation of neurosymbolic frameworks. Unifying ideas in the field and building flexible frameworks by incorporating strengths in every facet will ease the learning curve associated with NeSy systems and improve standardization. Future NeSy frameworks should aim to provide flexible implementation, a user-friendly interface, improve scalability, and develop seamless integrations with foundation models. The advent of next-generation LLMs/VLMs provides promising solutions to longstanding knowledge engineering challenges, fostering more effective and scalable integration of symbolic representations and advancing research in neurosymbolic AI.

## Acknowledgments

This project is partially supported by the Office of Naval Research (ONR) grant N00014-23-1-2417. Any opinions, findings, conclusions, or recommendations expressed in this material are those of the authors and do not necessarily reflect the views of the Office of Naval Research. We thank Danial Kamali for his help in early stages of drafting and Uzair Mohammad for his editing and suggestions for Figure 1.

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

## Appendix A. Example Task

NeSy frameworks formulate problems differently based on their implementation and setup. In **DomiKnowS**, the problem is reformulated as a logical constraint solving problem. This is done by representing the domain as a graph $G(V, E)$, where the nodes are the concepts in the domain and the edges are the relationships between them. Each node can have properties. The final logical constraint formulation is done using the defined concepts. In **DeepProbLog**, the problem is viewed as a combination of perception and reasoning, where the perception is the neural component that is fed as neural predicates into the reasoning component made of probabilistic logic programming with ProbLog. To solve a problem in DeepProbLog, the problem needs to be conceptualized as a separation of the neural and logical reasoning components. In **Scallop**, similar to DeepProbLog, the problem is viewed as a combination of the neural and the symbolic component. In **LEFT**, the problem is limited to the visual question answering domain. Here, the neural model is composed of feature extractors, a classifier for objects and relations into concepts, and a first-order logic program generator when given the question. In this section, we will take a look at how the problem formulation looks like in each of these frameworks for a common task. **Note** that we exclude LEFT for this due to the domain-specific nature of the framework.

### A.1. MNIST Sum

The MNIST Sum task is an extension of the classic MNIST handwritten digit recognition task (Lecun et al., 1998) where on being given two images of digits, the task is to output their sum that is a whole number. The training example consists of the two images of the digits and the ground-truth label of their sum. The individual labels of the digits are not available for training.

### A.1.1. DOMIKNOWS

**Problem Specification.** DomiKnowS formulates the problem using graph representations of concepts, relations, and logic. For performing the MNIST Sum task in DomiKnowS, the first concept defined is *image* concept representing visual information. The *digit* concept, a subclass of image, is introduced to represent the output class, ranging from 0 to 9. To establish relationships between digit images, the *image pair* concept is defined as an edge connecting two digit concepts. The sum concept is then introduced under image pair to represent the summation of the two digit concepts and the ground-truth output of the program. For this task, three constraints are defined. The first two constraints utilize *exactL* to ensure that the predicted digit and sum values belong to only one valid class. Another constraint enforces that the expected sum value matches the sum of the two digit predictions. This is implemented using *ifL* constraints, which verify whether the predicted digits form one of the possible solutions for a valid sum. If multiple solutions exist, the *orL* constraint ensures that at least one of the answers corresponds to the predicted digits. Code for defining the graph concept and constraints can be found in Listing 1.

```
with Graph(name='global') as graph:
    image_batch = Concept(name='image_batch')
    image = Concept(name='image')

    image_contains, = image_batch.contains(image)

    # digit classes 0-9
    digit = image(name='digits',
                  ConceptClass=EnumConcept,
                  values=digits)

    image_pair = Concept(name='pair')
    pair_d0, pair_d1 = image_pair.has_a(digit0=image, digit1=
        image)

    # sum value classes 0-18
    s = image_pair(name='summations',
                   ConceptClass=EnumConcept,
                   values=summations)

    exactL(*[digit.__getattr__(d) for d in digits])
    exactL(*[s.__getattr__(d) for d in summations])

    #fixedL(s)
    FIXED = True
    fixedL(s("x", eqL(image_pair, "summationEquality", {True})
        ), active = FIXED)

    for sum_val in range(config.summationRange):
        sum_combinations = []

        sum_nm = summations[sum_val]

        for d0_val in range(sum_val + 1):
            d1_val = sum_val - d0_val

            if d0_val >= len(digits) or d1_val >= len(digits):
                continue

            d0_nm = digits[d0_val]
            d1_nm = digits[d1_val]

            # for each combination of digits that sum to
                sum_val add constraint to list
```

```
            sum_combinations.append(andL(getattr(digit, d0_nm)
                (path=('x', pair_d0)),
                                            getattr(digit, d1_nm)
                                                (path=('x',
                                                pair_d1))
                                            ))

        print(sum_val, '-', sum_combinations)

        # if the given summation value is some value, then the
            digits must be one of a set of
        # digit pairs that add to that value
        # i.e. if sum val = s, d0 = 0 and d1 = s or d0 = 1 and
            d1 = s-1 ...
        # e.g. if sum val = 1, d0 = 0 and d1 = 1 or d0 = 1 and
            d0 = 0
        if len(sum_combinations) == 1:
            ifL(
                getattr(s, sum_nm)('x'),
                sum_combinations[0]
            )
        else:
            ifL(
                getattr(s, sum_nm)('x'),
                orL(*sum_combinations)
            )
```

Listing 1: Python Code for full graph of MNIST sum implemented in DomiKnowS including logical constraints

**Neural Modeling.** The model declaration comprises standard neural modeling components, including data loading, pre-processing, neural network definition, and loss function specification. The process begins with the *ReaderSensor*, which reads the input image. Next, a relation concept is defined using another sensor, *JointSensor*, to establish connections between images. The module learner is then employed to generate an initial prediction for the digit concept, which is subsequently passed to another sensor, *FunctionalSensor*, to compute the sum of two images. The associated code is provided in Listing 2.

```
class Net(torch.nn.Module):
    def __init__(self):
        super().__init__()

        self.conv1 = nn.Conv2d(1, 6, 5)
        self.conv2 = nn.Conv2d(6, 16, 5)
```

```python
        self.pool = nn.MaxPool2d(2, 2)

        self.lin1 = nn.Linear(256, 128)
        self.lin2 = nn.Linear(128, 10)

        self.relu = nn.ReLU()

        self.drop = nn.Dropout(p=0.2)

        self.norm = nn.LayerNorm(256)

    def forward(self, x):
        x = torch.squeeze(x, dim=0)

        x = x.reshape(2, 1, 28, 28)

        x = self.conv1(x)
        x = self.relu(x)
        x = self.pool(x)

        x = self.conv2(x)
        x = self.relu(x)
        x = self.pool(x)

        x = x.reshape(2, -1)

        x = self.norm(x)

        x = self.lin1(x)
        x = self.relu(x)

        x = self.drop(x)

        y_digit = self.lin2(x)

        return y_digit

class SumLayer(torch.nn.Module):
    def __init__(self):
        super().__init__()

        self.lin1 = nn.Linear(20, 64)
        self.lin2 = nn.Linear(64, 19)
```

```python
        self.relu = nn.ReLU()

    def forward(self, digits, do_time=True):
        digit0 = torch.unsqueeze(digits[0, :], dim=0)
        digit1 = torch.unsqueeze(digits[1, :], dim=0)

        x = torch.cat((digit0, digit1), dim=1)

        x = self.lin1(x)
        x = self.relu(x)

        y_sum = self.lin2(x)

        #return torch.zeros((1, 19), requires_grad=True)
        return y_sum

class SumLayerExplicit(torch.nn.Module):
    def __init__(self, device='cpu'):
        super().__init__()
        self.device = device

    def forward(self, digits, do_time=True):
        digit0 = torch.unsqueeze(digits[0, :], dim=0)
        digit1 = torch.unsqueeze(digits[1, :], dim=0)

        digit0 = F.softmax(digit0, dim=1)
        digit1 = F.softmax(digit1, dim=1)

        digit0 = torch.reshape(digit0, (10, 1))
        digit1 = torch.reshape(digit1, (1, 10))
        d = torch.matmul(digit0, digit1)
        d = d.repeat(1, 1, 1, 1)
        f = torch.flip(torch.eye(10), dims=(0,)).repeat(1, 1,
            1, 1)
        conv_diag_sums = F.conv2d(d, f.to(self.device),
            padding=(9, 0), groups=1)[..., 0]

        out = torch.squeeze(conv_diag_sums, dim=0)
        return out

class NBSoftCrossEntropyLoss(NBCrossEntropyLoss):
    def __init__(self, prior_weight=1.0, *args, **kwargs):
        super().__init__(*args, **kwargs)
```

```python
        self.prior_weight = prior_weight

    def forward(self, input, target, *args, **kwargs):
        if target.dim() == 1:
            return super().forward(input, target, *args, **
                kwargs)

        epsilon = 1e-5
        input = input.view(-1, input.shape[-1])
        input = input.clamp(min=epsilon, max=1-epsilon)

        logprobs = F.log_softmax(input, dim=1)
        return self.prior_weight * -(target * logprobs).sum()
            / input.shape[0]

class NBSoftCrossEntropyIMLoss(BCEWithLogitsIMLoss):
    def __init__(self, prior_weight=1.0, *args, **kwargs):
        super().__init__(*args, **kwargs)

        self.prior_weight = prior_weight

    def forward(self, input, inference, target, weight=None):
        if target.dim() == 1:
            num_classes = input.shape[-1]
            target = target.to(dtype=torch.long)
            target = F.one_hot(target, num_classes=num_classes
                )

            return super().forward(input, inference, target,
                weight=weight)

        return super().forward(input, inference, target,
            weight=weight) * self.prior_weight

def print_and_output(x, f=lambda x: x.shape, do_print=False):
    if do_print:
        print(prefix + str(f(x)))
    return x

def build_program(sum_setting=None, digit_labels=False, device
    ='cpu', use_fixedL=True, test=False):
```

```python
image['pixels'] = ReaderSensor(keyword='pixels')

def make_batch(pixel):
    return pixel.flatten().unsqueeze(0), torch.ones((1,
        len(pixel)))
image_batch['pixels', image_contains.reversed] =
    JointSensor(image['pixels'], forward=make_batch)

image['logits'] = ModuleLearner('pixels', module=Net())

def make_pairs(*inputs):
    return torch.tensor([[True, False]]), torch.tensor([[
        False, True]])

image_pair[pair_d0.reversed, pair_d1.reversed] =
    JointSensor(image['pixels'], forward=make_pairs)

image_pair['summation_label'] = ReaderSensor(keyword='
    summation')

image['digit_label'] = ReaderSensor(keyword='digit')

image[digit] = FunctionalSensor('logits', forward=lambda x
    : x)

if digit_labels:
    image[digit] = FunctionalSensor('digit_label', forward
        =lambda x: x, label=True)

if use_fixedL and test:
    # during test time, set model output to be the
        summation label
    def manual_fixedL(s):
        res = torch.zeros((1, 19))
        res[0, s] = 1
        return res

    image_pair[s] = FunctionalSensor('summation_label',
        forward=manual_fixedL)
else:
    if sum_setting == 'explicit':
        image_pair[s] = ModuleLearner(image['logits'],
            module=SumLayerExplicit(device=device))
    elif sum_setting == 'baseline':
```

```
        image_pair[s] = ModuleLearner(image['logits'],
            module=SumLayer())
    else:
        image_pair[s] = FunctionalSensor(forward=lambda:
            torch.ones(1, config.summationRange))  # dummy
            values to populate

if use_fixedL:
    image_pair[s] = ReaderSensor(keyword='summation',
        label=True)
    image_pair['summationEquality'] = FunctionalSensor(
        forward=lambda: torch.ones(1,1))

return graph, image, image_pair, image_batch
```

Listing 2: MNIST Sum code for DomiKnowS framework to run this task

## A.1.2. DeepProbLog

**Problem Specification.** DeepProbLog formulates a problem regarding probabilistic facts, neural facts, and neural annotated disjunctions (nAD). In the MNIST Sum task, the fact $X$ is defined to represent the input image. A neural network function is then introduced to map $X$ to its corresponding digit, denoted as digit$(X,Y)$. To enforce constraints about the summation and the ground-truth sum, a function is defined to compute the sum of two digits. Code for this part is shown in Listing 3.

```
nn(m_digit, [X], Y, [0.....9]) :: digit(X,Y).
addition(X,Y,Z) :- digit(X,X2), digit(Y,Y2), Z is X2+Y2.
```

Listing 3: Facts and Rules in DeepProbLog

**Neural Modeling.** The neural modeling follows a standard neural network setup, such as a CNN-based classifier. It is preceded by data loading and pre-processing, which are performed separately from the ProbLog program. Thus, the neural model used in Deep-ProbLog can be initialized independently of the DeepProbLog model. Once the neural model is initialized, the framework passes it along with a probabilistic program as input. The probabilistic program consists of facts and rules, similar to the code in Listing 3. Details of the DeepProbLog modeling code can be found in Listing 4.

```
class Model(object):
    def __init__(
        self,
        program_string: Union[str, os.PathLike],
        networks: Collection[Network],
        embeddings: Optional[TermEmbedder] = None,
        load: bool = True,
    ):
        """
```

```
    :param program_string: A string representing a
        DeepProbLog program or the path to a file
        containing a program.
    :param networks: A collection of networks that will be
        used to evaluate the neural predicates.
    :param embeddings: A TermEmbedder used to embed Terms
        in the program.
    :param load: If true, then it will attempt to load the
        program from 'program_string',
     else, it will consider program_string to be the
        program itself.
    """
    self.networks = dict()
    if load:
        self.program: LogicProgram = PrologFile(str(
            program_string))
    else:
        self.program: LogicProgram = PrologString(
            program_string)
    self.parameters = []
    self.parameter_groups = []
    self._extract_parameters()
    for network in networks:
        self.networks[network.name] = network
        network.model = self
    self.solver: Optional[Solver] = None
    self.eval_mode = False
    self.embeddings = embeddings
    self.tensor_sources = dict()
    self.optimizer = Optimizer(self)

def get_embedding(self, term: Term):
    return self.embeddings.get_embedding(term)

def evaluate_nn(self, to_evaluate: List[Tuple[Term, Term
]]):
    """
    :param to_evaluate: List of neural predicates to
        evaluate
    :return: A dictionary with the elements of to_evaluate
        as keys, and the output of the NN as values.
    """
    result = dict()
    evaluations = defaultdict(list)
```

```python
        # Group inputs per net to send in batch
        for net_name, inputs in to_evaluate:
            net = self.networks[str(net_name)]
            if net.det:
                tensor_name = Term("nn", net_name, inputs)
                if tensor_name not in self.solver.engine.
                    tensor_store:
                    evaluations[net_name].append(inputs)
            else:
                if inputs in net.cache:
                    result[(net_name, inputs)] = net.cache[
                        inputs]
                    del net.cache[inputs]
                else:
                    evaluations[net_name].append(inputs)
        for net in evaluations:
            network = self.networks[str(net)]
            out = network([term2list(x, False) for x in
                evaluations[net]])
            for i, k in enumerate(evaluations[net]):
                if network.det:
                    tensor_name = Term("nn", net, k)
                    self.solver.engine.tensor_store.store(out[
                        i], tensor_name)
                else:
                    result[(net, k)] = out[i]
        return result

    def set_engine(self, engine: Engine, **kwargs):
        """
        Initializes the solver of this model with the given
            engine and additional arguments.
        :param engine: The engine that will be used to ground
            queries in this model.
        :param kwargs: Additional arguments passed to the
            solver.
        :return:
        """
        self.solver = Solver(self, engine, **kwargs)
        register_tensor_predicates(engine)

    def solve(self, batch: Sequence[Query]) -> List[Result]:
        return self.solver.solve(batch)

    def ground_dataset(self, dataset: Dataset):
```

```python
        total_time = 0
        compile_times = []
        ground_times = []
        for q in dataset.to_queries():
            start = time.time()
            result = self.solver.cache.get(q)
            total_time += time.time() - start
            if not result.from_cache:
                compile_times.append(result.compile_time)
                ground_times.append(result.ground_time)
        return {
            "total_time": total_time,
            "ground_times": ground_times,
            "compile_times": compile_times,
        }

def save_state(self, filename: Union[str, PathLike, IO[
    bytes]], complete=False):
    """
    Saves the state of this model to a zip file with the
        given filename. This only includes the
        probabilistic
        parameters and all parameters of the neural
            networks, but not the model architecture or
            neural architectures
    :param filename: The filename to save the model to.
    :param complete: If true, save neural networks with
        information needed to resume training.
    :return:
    """
    check_path(filename)
    with ZipFile(filename, "w") as zipf:
        with zipf.open("parameters", "w") as f:
            pickle.dump(self.parameters, f)
        for n in self.networks:
            with zipf.open(n, "w") as f:
                self.networks[n].save(f, complete=complete
                    )

def load_state(self, filename: Union[str, PathLike, IO[
    bytes]]):
    """
    Restore the state of this model from the given
        filename. This only includes the probabilistic
        parameters
    """
```

```
            and all parameters of the neural networks, but not
                the model architecture or neural architectures
                    .
        :param filename: The filename to restore the model
            from.
        :return:
        """
        with ZipFile(filename) as zipf:
            with zipf.open("parameters") as f:
                self.parameters = pickle.load(f)
            for n in self.networks:
                with zipf.open(n) as f:
                    self.networks[n].load(BytesIO(f.read()))

    def eval(self):
        """
        Set the mode of all networks in the model to eval.
        """
        self.eval_mode = True
        for n in self.networks:
            self.networks[n].eval()
        self.solver.engine.eval()

    def train(self):
        """
        Set the mode of all networks in the model to train.
        :return:
        """
        self.eval_mode = False
        for n in self.networks:
            self.networks[n].train()
        self.solver.engine.train()

    def register_foreign(
        self, func: Callable, function_name: str, arity_in:
            int, arity_out: int
    ):
        self.solver.engine.register_foreign(func,
            function_name, arity_in, arity_out)

    def __str__(self):
        return "\n".join(str(line) for line in self.program)

    def get_tensor(self, term: Term) -> torch.Tensor:
        """
```

```python
    :param term: A term of the form tensor(_).
    If the tensor is of the form tensor(a(*args)), then it
        well look into tensor source a.
    :return:   Returns the stored tensor identifier by the
        term.
    """
    if len(term.args) > 0 and term.args[0].functor in self
        .tensor_sources:
        return self.tensor_sources[term.args[0].functor][
            term.args[0].args]
    return self.solver.get_tensor(term)

def store_tensor(self, tensor: torch.Tensor) -> Term:
    """
    Stores a tensor in the tensor store and returns and
        identifier.
    :param tensor: The tensor to store.
    :return: The Term that is the identifier by which this
        tensor can be uniquely identified in the logic.
    """
    return Term("tensor", Constant(self.solver.engine.
        tensor_store.store(tensor)))

def add_tensor_source(
    self, name: str, source: Union[ImageDataset, Mapping[
        Any, torch.Tensor]]
):
    """
    Adds a named tensor source to the model.
    :param name: The name of the added tensor source.
    :param source: The tensor source to add
    :return:
    """
    self.tensor_sources[name] = source

def get_hyperparameters(self) -> dict:
    """
    Recursively build a dictionary containing the most
        important hyperparameters in the model.
    :return: A dictionary that contains the values of the
        most important hyperparameters of the model.
    """
    parameters = dict()
    parameters["solver"] = (
```

```python
            None if self.solver is None else self.solver.
                get_hyperparameters()
        )
        parameters["networks"] = [
            self.networks[network].get_hyperparameters() for
                network in self.networks
        ]
        parameters["program"] = self.program.to_prolog()
        return parameters

    def hyperparameters_to_file(self, filename):
        """
        Write the output of the get_hyperparameter() method in
            JSON format to a file.
        :param filename: The path to write the hyperparameters
            to.
        :return:
        """
        with open(filename, "w") as f:
            f.write(json.dumps(self.get_hyperparameters()))

    def _extract_parameters(self):
        translated = SimpleProgram()
        for n in self.program:
            if type(n) is Term:
                if (
                    n.probability is not None
                    and type(n.probability) is Term
                    and n.probability.functor == "t"
                ):
                    i = self._add_parameter(n.probability.args
                        [0])
                    p = n.probability.with_args(Constant(i))
                    n = n.with_probability(p)
                translated.add_statement(n)
            elif type(n) is Clause:
                if (
                    n.head.probability is not None
                    and type(n.head.probability) is Term
                    and n.head.probability.functor == "t"
                ):
                    i = self._add_parameter(n.head.probability
                        .args[0])
                    p = n.head.probability.with_args(Constant(
                        i))
```

```
                    head = n.head.with_probability(p)
                    n = Clause(head, n.body)
                translated.add_statement(n)
            elif type(n) is Or:
                new_list = []
                new_group = []
                for x in n.to_list():
                    if (
                        x.probability is not None
                        and type(x.probability) is Term
                        and x.probability.functor == "t"
                    ):
                        i = self._add_parameter(x.probability.
                            args[0])
                        new_group.append(i)
                        p = x.probability.with_args(Constant(i
                            ))
                        new_list.append(x.with_probability(p))
                    else:
                        new_list.append(x)
                if len(new_group) > 0:
                    self.parameter_groups.append(new_group)
                n = Or.from_list(new_list)
                translated.add_statement(n)
            else:
                translated.add_statement(n)
        self.program = translated

    def _add_parameter(self, val: Constant):
        i = len(self.parameters)
        try:
            val = float(val)
        except InstantiationError:
            val = random()
        self.parameters.append(val)
        return i
```

Listing 4: Example of code for neural model in DeepProbLog for MNIST Sum task.

A.1.3. SCALLOP

**Problem Specification.** Scallop formulates the problem in terms of relations, values, and (Horn) rules derived from Datalog. As discussed earlier, the concepts and constraints defined in this framework are similar to those in DeepProbLog. However, these rules can be directly embedded into a Scallop program through its API. The process begins by establishing the concepts *digit1* and *digit2* to represent the digit values of two given images. Constraints

are then defined based on the summation of these two values, which must be equal to the
*sum_2* concept, serving as the ground truth for this task. The code in Listing 5 provides a
portion of the implementation for defining these concepts and constraints.

```
self.scl_ctx.add_relation(``digit_1'', int, input_mapping=list
    (range(10)))
self.scl_ctx.add_relation(``digit_2'', int, input_mapping=list
    (range(10)))
self.scl_ctx.add_rule(``sum_2(a + b) :- digit_1(a), digit_2(b)
    '')
self.sum_2 = self.scl_ctx.forward_function(``sum_2'',
    output_mapping=[(i,) for i in range(19)])
```

Listing 5: Code for defining the concept and constraints in Scallop framework for MNIST
sum task

**Neural Modeling.** Unlike DeepProbLog, the neural modeling is integrated with Scallop's
relation and rule declaration. The neural modeling remains a standard neural network.
Details of the modeling code can be found in Listing 6.

```
mnist_img_transform = torchvision.transforms.Compose([
  torchvision.transforms.ToTensor(),
  torchvision.transforms.Normalize(
    (0.1307,), (0.3081,)
  )
])

class MNISTSum2Dataset(torch.utils.data.Dataset):
  def __init__(
    self,
    root: str,
    train: bool = True,
    transform: Optional[Callable] = None,
    target_transform: Optional[Callable] = None,
    download: bool = False,
  ):
    # Contains a MNIST dataset
    self.mnist_dataset = torchvision.datasets.MNIST(
      root,
      train=train,
      transform=transform,
      target_transform=target_transform,
      download=download,
    )
    self.index_map = list(range(len(self.mnist_dataset)))
    random.shuffle(self.index_map)
```

```python
  def __len__(self):
    return int(len(self.mnist_dataset) / 2)

  def __getitem__(self, idx):
    # Get two data points
    (a_img, a_digit) = self.mnist_dataset[self.index_map[idx *
        2]]
    (b_img, b_digit) = self.mnist_dataset[self.index_map[idx *
        2 + 1]]

    # Each data has two images and the GT is the sum of two
        digits
    return (a_img, b_img, a_digit + b_digit)

  @staticmethod
  def collate_fn(batch):
    a_imgs = torch.stack([item[0] for item in batch])
    b_imgs = torch.stack([item[1] for item in batch])
    digits = torch.stack([torch.tensor(item[2]).long() for
        item in batch])
    return ((a_imgs, b_imgs), digits)

def mnist_sum_2_loader(data_dir, batch_size_train,
    batch_size_test):
  train_loader = torch.utils.data.DataLoader(
    MNISTSum2Dataset(
      data_dir,
      train=True,
      download=True,
      transform=mnist_img_transform,
    ),
    collate_fn=MNISTSum2Dataset.collate_fn,
    batch_size=batch_size_train,
    shuffle=True
  )

  test_loader = torch.utils.data.DataLoader(
    MNISTSum2Dataset(
      data_dir,
      train=False,
      download=True,
      transform=mnist_img_transform,
    ),
```

```
        collate_fn=MNISTSum2Dataset.collate_fn ,
        batch_size=batch_size_test ,
        shuffle=True
    )

    return train_loader , test_loader

class MNISTNet(nn.Module):
    def __init__(self):
        super(MNISTNet, self).__init__()
        self.conv1 = nn.Conv2d(1, 32, kernel_size=5)
        self.conv2 = nn.Conv2d(32, 64, kernel_size=5)
        self.fc1 = nn.Linear(1024, 1024)
        self.fc2 = nn.Linear(1024, 10)

    def forward(self, x):
        x = F.max_pool2d(self.conv1(x), 2)
        x = F.max_pool2d(self.conv2(x), 2)
        x = x.view(-1, 1024)
        x = F.relu(self.fc1(x))
        x = F.dropout(x, p = 0.5, training=self.training)
        x = self.fc2(x)
        return F.softmax(x, dim=1)

class MNISTSum2Net(nn.Module):
    def __init__(self, provenance, k):
        super(MNISTSum2Net, self).__init__()

        # MNIST Digit Recognition Network
        self.mnist_net = MNISTNet()

        # Scallop Context
        self.scl_ctx = scallopy.ScallopContext(provenance=
            provenance, k=k)
        self.scl_ctx.add_relation("digit_1", int, input_mapping=
            list(range(10)))
        self.scl_ctx.add_relation("digit_2", int, input_mapping=
            list(range(10)))
        self.scl_ctx.add_rule("sum_2(a + b) :- digit_1(a), digit_2
            (b)")

        # The `sum_2` logical reasoning module
```

```
    self.sum_2 = self.scl_ctx.forward_function("sum_2",
        output_mapping=[(i,) for i in range(19)], jit=args.jit,
        dispatch=args.dispatch)

  def forward(self, x: Tuple[torch.Tensor, torch.Tensor]):
    (a_imgs, b_imgs) = x

    # First recognize the two digits
    a_distrs = self.mnist_net(a_imgs) # Tensor 64 x 10
    b_distrs = self.mnist_net(b_imgs) # Tensor 64 x 10

    # Then execute the reasoning module; the result is a size
        19 tensor
    return self.sum_2(digit_1=a_distrs, digit_2=b_distrs) #
        Tensor 64 x 19

def bce_loss(output, ground_truth):
  (_, dim) = output.shape
  gt = torch.stack([torch.tensor([1.0 if i == t else 0.0 for i
      in range(dim)]) for t in ground_truth])
  return F.binary_cross_entropy(output, gt)

def nll_loss(output, ground_truth):
  return F.nll_loss(output, ground_truth)

class Trainer():
  def __init__(self, train_loader, test_loader, model_dir,
      learning_rate, loss, k, provenance):
    self.model_dir = model_dir
    self.network = MNISTSum2Net(provenance, k)
    self.optimizer = optim.Adam(self.network.parameters(), lr=
        learning_rate)
    self.train_loader = train_loader
    self.test_loader = test_loader
    self.best_loss = 10000000000
    if loss == "nll":
      self.loss = nll_loss
    elif loss == "bce":
      self.loss = bce_loss
    else:
      raise Exception(f"Unknown loss function `{loss}`")
```

```python
def train_epoch(self, epoch):
    self.network.train()
    iter = tqdm(self.train_loader, total=len(self.train_loader
        ))
    for (data, target) in iter:
        self.optimizer.zero_grad()
        output = self.network(data)
        loss = self.loss(output, target)
        loss.backward()
        self.optimizer.step()
        iter.set_description(f"[Train Epoch {epoch}] Loss: {loss
            .item():.4f}")

def test_epoch(self, epoch):
    self.network.eval()
    num_items = len(self.test_loader.dataset)
    test_loss = 0
    correct = 0
    with torch.no_grad():
        iter = tqdm(self.test_loader, total=len(self.test_loader
            ))
        for (data, target) in iter:
            output = self.network(data)
            test_loss += self.loss(output, target).item()
            pred = output.data.max(1, keepdim=True)[1]
            correct += pred.eq(target.data.view_as(pred)).sum()
            perc = 100. * correct / num_items
            iter.set_description(f"[Test Epoch {epoch}] Total loss
                : {test_loss:.4f}, Accuracy: {correct}/{num_items} 
                ({perc:.2f}%)")
        if test_loss < self.best_loss:
            self.best_loss = test_loss
            torch.save(self.network, os.path.join(model_dir, "
                sum_2_best.pt"))

def train(self, n_epochs):
    self.test_epoch(0)
    for epoch in range(1, n_epochs + 1):
        self.train_epoch(epoch)
        self.test_epoch(epoch)
```

Listing 6: Example of code for neural model in Scallop for MNIST Sum task.

