# OpenReview forum: "Toward a Clearer Characterization of Neuro-Symbolic Frameworks: A Brief Comparative Analysis"
_nesyconf.org/NeSy/2025/Conference_Phase_2 — NeSy 2025 - Phase 2 Poster_

### Official Review · Reviewer_mM7V · 2025-07-04

**Rating:** 5
**Confidence:** 3

**Review:**

This paper compares three NeSy methods, DeepProbLog, Scallop and DomiKnowS through the lens of their symbolic knowledge representation, neural model architecture, computational efficiency, and their ability to perform model composition. The authors then discuss several aspects of NeSy methods and provides some future directions.

Pros:
1. Given recent attention on NeSy methods, this paper does an important work of providing new researchers and potential practitioners with information about this field.
2. The paper does a good job of demonstrating the pros and cons of the three methods discussed in the paper.

Cons:
1. The paper does not properly define what constitutes a NeSy framework. NeSy methods are well and truly understood, however since the author made distinctions between methods that deal only with specific tasks and methods that are task independent, it's unclear if the task independence is what makes a NeSy method a framework.
2. The authors made comparisons across several dimensions, but did not provide any justification to use those dimensions. The motivation for the paper (as in the abstract) is stated to be learning curve, user-friendly tools, libraries and unifying framework, but none of the chosen dimensions are related to them. Furthermore, if we accept that the authors are defining frameworks as task independent, then under what criteria should we consider a framework over another? The authors' chosen dimensions don't seem to support that either.
3. The paper title includes "What's Next?" but the paper only spends roughly half a page discussing future possibilities (out of 10 pages). The authors would do well to either remove that from the title or expanding the section discussing future work.
4. The paper is also not very consistent with its comparison dimensions. In the section heading, we see knowledge representation, neural architecture, model composition, interplay and LLM. However in Table 1, it is knowledge representation, model composition, efficiency and LLM. In the discussion section, it is knowledge representation, model composition, classification and usability. This inconsistency can confuse potential readers.

Overall, this paper provides an important source of information to potential practitioners and incoming researchers, but lacks rigor in defining and justifying certain terms used (e.g. framework and dimensions used for comparison). Improving the consistency in the overall organization would lead to a better paper.

**Anonymity:**

Remain anonymous

---

### Official Review · Reviewer_DqqX · 2025-07-08
**Review of : A Comparative Analysis of NeSy Frameworks and What's Next?**

**Rating:** 7
**Confidence:** 3

**Review:**

The paper states that a majority of the
NeSy literature focuses on tasks and proposes task-specific methodologies instead of pro-viding task-independent frameworks for others to leverage for problem solving.

This overview looks into three "task-independent" approaches: DeepProbLog, Scallop and DomiKnowS, identifying the "challenges within each facet that lay the foundation for identifying the expressivity of each framework in solving a variety of problems."

The paper covers several aspects of Neurosymbolic AI, including
"[the] Illustration of the main components of existing NeSy frameworks", [a] "Comparison of frameworks across identified comparative facets"
[the] "Identification of requirements for the next generation of NeSy frameworks and systems, building upon the drawbacks of the
current systems and the existing characterizations of possible interplays between the neural
and symbolic systems."

The authors then proceed in the next sections by charazterizing each of the chose framerworks by:
1) The symbolic knowledge and data representation language;
2) The representation and flexibility of Neural Modeling;
3) Model Composition;
4) The interplay between symbolic and sub-symbolic systems;
5) Their usage of LLMs.

Their analysis concludes by sketching a comparative analysis of the three chosen NeSy approaches (DeepProbLog, Scallop, and DomiKnowS).
By focusing on aspects of the 5 criteria above, the authors
identify key challenges in each facet that "can guide us towards building the next generation of neurosymbolic frameworks and systems."
Overall the paper can be a useful contribution to the practice of
NeSy engineering. Although the writing could be improved to explain
engineering aspects in more detail, if the authors proceed in
providing associated appropriate documentation, this can be a useful contribution to the community.

**Anonymity:**

Remain anonymous

---

### Official Review · Reviewer_wHE4 · 2025-07-08
**Review for A Comparative Analysis of NeSy Frameworks and What's Next?**

**Rating:** 4
**Confidence:** 3

**Review:**

This paper presents a comparative analysis of (mainly 3) neuro-symbolic frameworks, aiming to characterise their technical aspects and use this case study to guide future research. The authors focus on what they refer to as "task-independent" frameworks, selecting DeepProbLog, Scallop, and DomiKnowS for the in-depth comparison.

Remarks:
1. The paper is organised based on the distinction between "task-specific" methodologies and "task-independent" frameworks.
While this can be useful, the paper could do more to formalise this distinction. As it stands, the selection of the three frameworks feels somewhat arbitrary.

2. Revising Scope and Title: The authors should either  (i) broaden the scope of the paper to include a wider range of NeSy methods in the comparison to match the intent from the title, or (ii) revise the title to more accurately reflect the paper's contribution as a case study limited to the three methods analysed.

3. The paper does not include any significant discussion about real-world applications of the frameworks it analyses (DeepProbLog, Scallop, and DomiKnowS). The entire comparative analysis is grounded in the MNIST Sum task, which is a synthetic  benchmark.
The broader NeSy field is actively being applied to real-world domains, see e.g. CCN+ [1], which can handle arbitrary requirements expressed in propositional logic, and can be applied to complex scenarios like autonomous driving.

4. Typo: "Whil" --> "While" (first sentence of Section 2)

**References**

[1] E. Giunchiglia‚ A. Tatomir‚ M. C. Stoian,T. Lukasiewicz. CCN+: A neuro-symbolic framework for deep learning with requirements. International Journal of Approximate Reasoning, 171, 109-124 (2024).

**Anonymity:**

Remain anonymous